# Specialist or Generalist? Instruction Tuning for Specific NLP Tasks

**Chufan Shi**[♠][*]**, Yixuan Su**[◇,♣]**, Cheng Yang**[♠]**, Yujiu Yang**[♠]**, Deng Cai**[♡†]
[♠]Tsinghua Shenzhen International Graduate School, Tsinghua University
[◇]Language Technology Lab, University of Cambridge      [♣]Cohere
[♡]Tencent AI Lab
{scf22,yangc21}@mails.tsinghua.edu.cn, yang.yujiu@sz.tsinghua.edu.cn
yixuan@cohere.com   thisisjcykcd@gmail.com

## Abstract

The potential of large language models (LLMs) to simultaneously perform a wide range of natural language processing (NLP) tasks has been the subject of extensive research. Although instruction tuning has proven to be a data-efficient method for transforming LLMs into such generalist models, their performance still lags behind specialist models trained exclusively for specific tasks. In this paper, we investigate whether incorporating broad-coverage generalist instruction tuning can contribute to building a specialist model. We hypothesize that its efficacy depends on task specificity and skill requirements. Our experiments assess four target tasks with distinct coverage levels, revealing that integrating generalist instruction tuning consistently enhances model performance when the task coverage is broad. The effect is particularly pronounced when the amount of task-specific training data is limited. Further investigation into three target tasks focusing on different capabilities demonstrates that generalist instruction tuning improves understanding and reasoning abilities. However, for tasks requiring factual knowledge, generalist data containing hallucinatory information may negatively affect the model's performance. Overall, our work provides a systematic guide for developing specialist models with general instruction tuning. Our code and other related resources can be found at https://github.com/DavidFanzz/Generalist_or_Specialist.

## 1 Introduction

The latest generation of large language models (LLMs), such as ChatGPT (OpenAI, 2022) and GPT4 (OpenAI, 2023), are often referred to as *generalist* models for their exceptional generalizability to perform various natural language processing (NLP) tasks. Recent studies (Taori et al.,

---

[*]This work was completed during an internship at Tencent AI Lab.
[†]Corresponding author.

2023; Zhou et al., 2023; Gudibande et al., 2023) suggest that (1) the foundation of their superior performance (i.e., knowledge and capabilities) is predominantly acquired during large-scale unsupervised pre-training; and (2) instruction tuning (Sanh et al., 2022; Wei et al., 2022a; Mishra et al., 2021; Ouyang et al., 2022) is an incredibly data-efficient method for unleashing the power of LLMs to complete realistic NLP tasks. However, under rigorous evaluation, the performance of those instruction-following generalist models often falls short compared to traditional task-specific specialist models (Jiao et al., 2023b; Qin et al., 2023; Fang et al., 2023; Liu et al., 2023). Recently, there has also been a growing trend towards developing specialist models using instruction tuning (Jiao et al., 2023a; Wang et al., 2023b; Zhang et al., 2023; Cheng et al., 2023; Wang et al., 2023a).

In this paper, we study how to better harness the power of LLM for specific NLP tasks using instruction tuning. Our research is motivated by the existence of various broad-coverage general-purpose instruction-following datasets (Taori et al., 2023; Peng et al., 2023; Labs, 2023; Xu et al., 2023; Zhou et al., 2023; Su et al., 2023b) and their surprising efficiency for turning LLMs into capable instruction-following generalists. For instance, Zhou et al. (2023) shows that merely one thousand supervised input-output pairs are necessary to build a competent generalist. In contrast to general-purpose instruction tuning, our preliminary experiments show that a sufficiently large set of task-specific data is still required for transforming an LLM into a superior specialist. This leads us to a pivotal research question: How to better unleash the power of LLMs for specific NLP tasks by marrying the best of two worlds? More specifically, can general-purpose instruction-following datasets aid in the transformation of an LLM into a specialist? If so, when and how?

We hypothesize the answers to the previous ques-

tions depend on (1) how specific the target task is; and (2) what skills the target task requires. To test this hypothesis, we first assess four target tasks with distinct levels of coverage. Our findings reveal that integrating general instruction tuning—that is, training with generalist data enhances the model's performance on specific NLP tasks with broad task coverage, particularly when the amount of task-specific training data is limited. To gain a deeper understanding of the improvements elicited by training with generalist data, we subsequently examine three target tasks that focus on distinct skill sets. Our results suggest that general instruction tuning improves the model's understanding and reasoning capabilities. However, when it comes to tasks that demand factual knowledge from the LLM, instructional data generated through self-instruct (Wang et al., 2022a) harms the model's performance due to the intrinsic hallucinations brought by such data creation approach.

In sum, to the best of our knowledge, our work is the first effort to present a systematic guide for building and improving specialist models with general instruction tuning.

## 2 Background: Instruction Tuning

In recent years, large language models (LLMs) have undergone rapid development and have dominated the field of natural language processing (NLP) (Radford et al., 2018; Devlin et al., 2019; Radford et al., 2019; Brown et al., 2020). Today's LLMs, such as ChatGPT (OpenAI, 2022) and GPT-4 (OpenAI, 2023), can perform complex and diverse tasks in the unified form of following natural language instructions. Generally, these models are trained in three separate stages: (1) large-scale unsupervised pre-training on raw text; and (2) instruction tuning via supervised learning (Sanh et al., 2022; Wei et al., 2022a; Mishra et al., 2021; Su and Collier, 2022; Su et al., 2023b); and (3) reinforcement learning from human feedback (Stiennon et al., 2020; Bai et al., 2022; Ouyang et al., 2022). Recent studies (Zhou et al., 2023; Gudibande et al., 2023) argued that almost all capabilities of LLMs are learned during unsupervised pre-training, and instruction tuning with a limited amount of supervised data is sufficient. However, this observation refers to the process of constructing general-purpose instruction-following models—generalists. In the following, we separately introduce broad-coverage "generalist" and task-specific

"specialist" instruction tuning.

**Generalist Instruction Tuning.** Early attempts on instruction tuning (Wang et al., 2022b; Sanh et al., 2022; Wei et al., 2022a; Chung et al., 2022, *inter alia*) transform a range of public NLP datasets into an instructional format, with a few manually crafted templates for each task. They then fine-tune an LLM on a portion of the transformed data and evaluate on another set of held-out tasks. Each work affirms that the model's generalization ability to unseen tasks improves when increasing the task and template diversity. However, template-based instructions are not sufficiently diverse for building a truly competent generalist (Ouyang et al., 2022). In contrast, state-of-the-art generalist models such as ChatGPT (OpenAI, 2022) are trained with proprietary instructions collected from real human users. In the pursuit to replicate the success of ChatGPT, various open-source broad-coverage instruction-tuning datasets are proposed. Some are gathered via crowd-sourcing (Labs, 2023; Zhou et al., 2023) while others use the outputs from strong proprietary models (Taori et al., 2023; Peng et al., 2023; Xu et al., 2023; Su et al., 2023a; Li et al., 2023) with techniques such as self-instruct (Wang et al., 2022a). Existing results suggest that these models can achieve near parity with proprietary models in various aspects (Chiang et al., 2023; Zhou et al., 2023; Taori et al., 2023).

**Specialist Instruction Tuning.** There is also an emerging trend to continue instruction tuning on specific NLP tasks, such as machine translation (Jiao et al., 2023a), information extraction (Wang et al., 2023b), medical QA (Wang et al., 2023a; Fleming et al., 2023), and writing-assistant (Zhang et al., 2023). These works typically transform existing task-specific datasets into the same instructional format as generalist instruction tuning and yield better model performance in specific tasks. Different from previous work, this study aims to provide a comprehensive and in-depth investigation of the role of generalist instruction data in specialist instruction tuning.

Our work is most related to the initial studies on the cross-task generalization of instruction tuning such as FLAN (Wei et al., 2022a). The differences between our work and previous work are: (1) we use broad-coverage generalist data, while they use template-based data; and (2) they focus on zero/few-shot performance on unseen tasks, while

we assume an adequate amount of task-specific training data is available.

## 3 Incorporating Specialist Training with Generalist Training

### 3.1 Data Collection

We sort the instruction-following data into two groups: (1) specialist data and (2) generalist data.

**Specialist data.** primarily originates from existing NLP datasets with a focus on particular tasks. To facilitate our research, we mainly utilize the SuperNI dataset (Wang et al., 2022b), a comprehensive benchmark containing 1,616 NLP datasets coupled with their respective natural language instructions, as the source of specialist data. The details are described in Section 4.1. We also leverage existing question answering datasets (Kwiatkowski et al., 2019; Berant et al., 2013; Joshi et al., 2017) , reading comprehension datasets (Lai et al., 2017) reasoning datasets (Bowman et al., 2015; Talmor et al., 2019; Ling et al., 2017) to evaluate different aspects of model skills, detailed in Section 5.1.

**Generalist data** is characterized by its extensive scope and diversity. For our research, we select two representative broad-coverage general-purpose datasets: GPT4-Instruct (Peng et al., 2023) and LIMA (Zhou et al., 2023). GPT4-Instruct (Peng et al., 2023) contains 52k unique instruction-response pairs, where the instructions are collected through self-instruct (Wang et al., 2022a) and the responses are generated by GPT-4 (OpenAI, 2023). LIMA (Zhou et al., 2023) consists of 1k carefully curated instruction-response pairs derived from human-authored community questions and answers. Notably, we emphasize that GPT4-Instruct serves as an example of generalist data synthesized by LLMs and LIMA represents another distinct example of generalist data written by humans.

**Unified Format.** We follow the template used in Stanford's Alpaca project (Taori et al., 2023) (See Appendix A). Each instance in the generalist and specialist data is transformed in a pair of {instruction, response}.

### 3.2 Training

**Specialist/Generalist Data Combination.** For each target task, we construct the training and test set with 50k and 5k instances, respectively. For target tasks that span over multiple datasets, we uniformly sample training/test instances from the corresponding datasets such that each dataset has an equal proportion. For generalist data, we consider the GPT4-Instruct and LIMA datasets as discussed above. We first train models on generalist data and then specialist data. We vary the amounts of specialist data across {2k, 4k, 6k, 8k, 10k} to study the effect of generalist data under different circumstances of data scarcity.

**Model and Training Details.** We conduct our experiments with the popular LLaMA 7B and 13B models (Touvron et al., 2023). For training on generalist data, we follow the original setups in the respective papers (Zhou et al., 2023; Taori et al., 2023). Specifically, for GPT4-Instruct, we train for 3 epochs with a batch size of 128, while for LIMA, we train for 15 epochs with a batch size of 64. In the subsequent specialist training phase, we train for 3 epochs with a batch size of 128. In both stages, we use the Adam optimizer (Kingma and Ba, 2015) with a learning rate of 2e-5 and utilize the standard language modeling objective:

$$\mathcal{L} = -\frac{1}{|\boldsymbol{y}|} \sum_{i=1}^{|\boldsymbol{y}|} \log p_\theta(y_i | \boldsymbol{x}, \boldsymbol{y}_{<i}), \qquad (1)$$

where $\theta$ denotes the model parameters and $\{\boldsymbol{x}, \boldsymbol{y}\}$ is an instruction-response pair.

## 4 Experiments I: The Coverage of the Target Tasks

### 4.1 Coverage Taxonomy

To assess our model's performance on a variety of target tasks with distinct levels of generality, we construct a hierarchy of four specialist tasks using the SuperNI dataset (Wang et al., 2022b). This taxonomy encompasses tasks with varying scopes of coverage, as detailed below.

**SuperNI** (multiple tasks, multiple formats). At the most comprehensive level, we incorporate all the *English* tasks from the SuperNI dataset, which encompasses a total of 756 datasets. Unlike LIMA and GPT4-Instruct, which accommodate a broad spectrum of user-oriented inquiries, the datasets in SuperNI focus on specific NLP tasks distilled from real-world demands. Therefore, we treat them as specialist data at the highest coverage level.

**Classification** (multiple tasks, single format). The tasks in SuperNI can be grouped based on

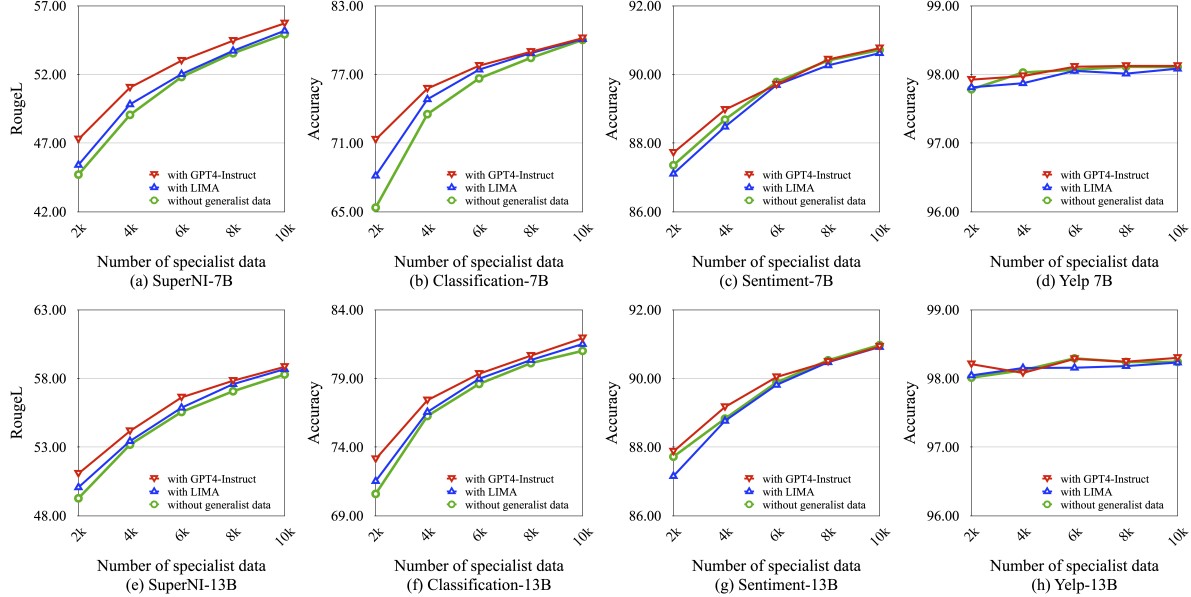

Figure 1: Comparison of models trained with different combinations of specialist and generalist data across different tasks. We report Rouge-L for SuperNI and accuracy for other levels.

their task types, such as classification, summarization, and question answering. For the second level, we focus on the classification subset. Specifically, we select 252 classification datasets. To measure the model's cross-task generalization capability, we allocate 223 datasets for training and reserve the remaining 29 datasets as held-out datasets for evaluation.

**Sentiment** (single tasks, multiple domains). The classification tasks selected above can be further categorized based on their specific topics, such as sentiment analysis, toxic language detection, commonsense categorization, and others. Among these, we designate 32 sentiment analysis datasets as the third level.

**Yelp** (single tasks, single domain). The sentiment analysis datasets mentioned above span various domains, such as movie and restaurant reviews. At the most fine-grained level, we choose the Yelp dataset (Zhang et al., 2015) as the representative task to evaluate the model's performance in a highly specialized domain.

### 4.2 Evaluation Setup

For the SuperNI level, we follow the same evaluation protocol as in Wang et al. (2022b) and report Rouge-L (Lin, 2004). For the decoding strategy, we adopt greedy search with a maximum generation

| Task Coverage | GPT4-Instruct | LIMA | specialist |
|---|---|---|---|
| SuperNI | 25.54 | 12.65 | 54.92 |
| Classification | 53.20 | 46.84 | 80.02 |
| Sentiment | 68.66 | 51.46 | 90.71 |
| Yelp | 91.68 | 65.52 | 98.11 |

Table 1: The performance of generalists and specialists on tasks of different coverage levels on LLaMA-7B. The specialists are trained with 10k task-specific instances. For SuperNI, the performance is measured by Rouge-L, while the others are measured by accuracy.

length of 512.[1] For the Classification, Sentiment, and Yelp levels, we follow previous studies (Brown et al., 2020; Sanh et al., 2022) and utilize a *classification with options* approach, where we prompt the model with a set of options and compute the likelihood of each option being the response. The one with the highest probability is taken as the model's prediction, and we report the model's accuracy.

### 4.3 Main Results

**Generalist models lag behind specialist models across all coverage levels.** We compare generalist models that are solely trained on generalist data (i.e., LIMA or GPT4-Instruct) to those specialist models that are solely trained on specialist data (the 10k training instances we collect for each cov-

---

[1]We leave the study on more advanced decoding methods (Holtzman et al., 2019; Su et al., 2022; Yang et al., 2023) as future work.

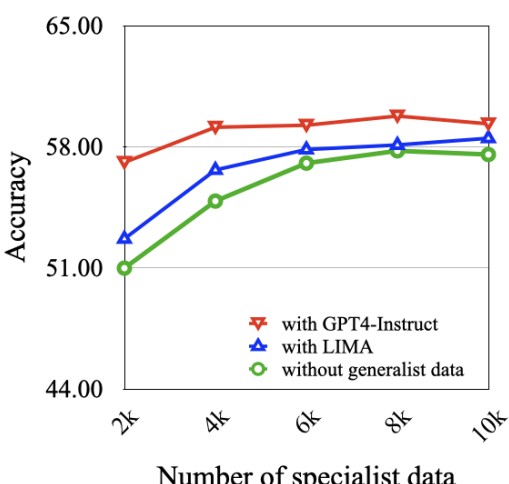

Figure 2: Results on held-out tasks (Classification) with LLaMA-7B.

erage level), using LLaMA-7B. From the results presented in Table 1, we can see that generalist models fall short in performance when compared to specialist models on all coverage levels. Notably, even as the coverage level becomes more encompassing, the performance gap between generalist models and specialist models does not shrink. For instance, on the most specific Yelp task, the specialist model outperforms the generalist model (GPT4-Instruct) by 6.43% absolute points. On the SuperNI task, the performance gap between the specialist and the generalist (GPT4-Instruct) is 29.38. These results validate the necessity of specialist tuning for specific NLP tasks.

**Transforming an LLM into a superior specialist demands a substantial amount of task-specific data.** Figure 1 depicts the performance of specialist models on different tasks with varying numbers of training data (from 2k to 10k). From the results, we see that, for tasks with broader coverage (e.g. SuperNI and Classification), the model's performance does not seem to converge with the 10k training instances. Even for narrow tasks such as Sentiment, at least 10k task-specific data is required to fully unlock the LLM's potential. These results reveal the data-hungry nature of building specialist models.

**Generalist data can improve specialist performance when the task coverage is broad.** Figure 1 also demonstrates that the inclusion of generalist data consistently results in performance improve-

ments for both SuperNI and Classification across LLaMA 7B and 13B models. On average across different settings of specialist data, the introduction of generalist data leads to an improvement of 0.96 for LLaMA-7B and 0.74 for LLaMA-13B on SuperNI tasks, while for Classification tasks, it results in an enhancement of 1.53% for LLaMA-7B and 0.82% for LLaMA-13B. It is also worth noting that LIMA only has 1k instances, but it can even help improve performance when the number of specialist data is $10\times$ larger. However, the results are the opposite for Sentiment and Yelp. For instance, the introduction of LIMA leads to a minor performance degeneration on Sentiment with 2k specialist data (a reduction of 0.25% for LLaMA-7B and 0.56% for LLaMA-13B). In the case of the Yelp task, the impact of including generalist data (both GPT4-Instruct and LIMA) appears to be minimal on the overall model performance.

**The performance gain is most evident when the amount of specialist data is limited.** We can see that the performance gap between specialists trained with and without generalist data shrinks as the amount of specialist data increases. For example, at the Classification level, when the specialist data comprises only 2k instances, the inclusion of GPT4-Instruct enhances LLaMA-7B's accuracy from 65.36% to 71.31% (+5.95%) and LLaMA-13B's accuracy from 70.59% to 73.13% (+2.54%). However, when the number of specialist data reaches 10k instances, the addition of GPT4-Instruct only leads to smaller improvements, from 80.02% to 80.17% (+0.15%) for LLaMA-7B, and from 81.01% to 81.93% (+0.92%) for LLaMA-13B, respectively.

**The performance gain is less pronounced when the model scale is larger.** As shown in Figure 1, when comparing the results of the 7B and 13B models, the trend of change in the effect of integrating generalist data is consistent for both models. However, it is worth noting that as the model scale is larger, the performance gain is less pronounced. Specifically, when the model scales up from 7B to 13B, the average improvement achieved by adding GPT4-Instruct on SuperNI decreases from 1.49 to 0.58, and the improvement in Classification reduces from 2.00% to 1.18%.

### 4.4 Further Analysis

For a deeper understanding of the impact of generalist data, here we present additional analyses.

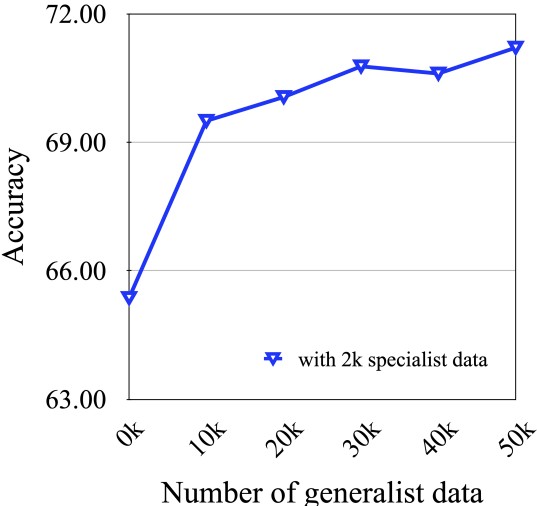

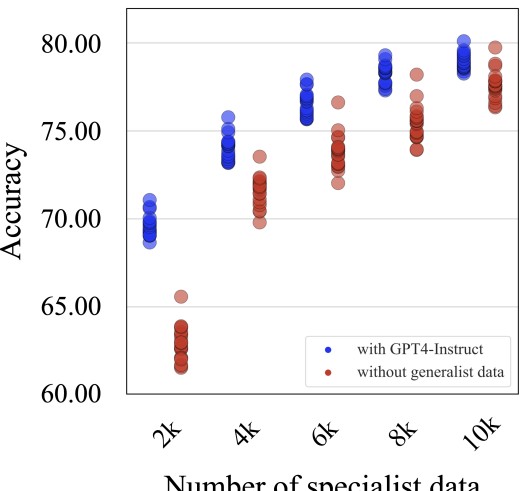

Figure 3: Results using different amounts of generalist data (Classification, 10k specialist data) with LLaMA-7B.

Figure 4: The results using different test instructions (Classification) with LLaMA-7B.

Unless otherwise specified, all experiments use LLaMA-7B as the foundation model.

**Cross-task Generalization.** For the Classification level, recall that we exclude some classification tasks when constructing the training data. These tasks can be used as hold-out tasks to examine the specialist's cross-task generalization ability. The results are shown in Figure 2. It can be observed that the accuracy on held-out tasks fluctuates in small ranges from 50.98% to 57.55% across different amounts of specialist data. However, upon incorporating LIMA, the average absolute accuracy improvement on the hold-out task increases by 2.70%, while adding GPT4-Instruct results in a 6.12% rise in absolute accuracy. This indicates that generalist data can greatly improve the cross-task generalization of specialist models.

**Number of Generalist Data.** To study the effect of the amount of generalist data, we additionally partition the GPT4-Instruct dataset into five random parts and test the model's performance when using different proportions of the dataset. The experiments are conducted at the Classification level with a fixed quantity of 10k specialist data. As shown in Figure 3, even with only 10k generalist data, the model's accuracy is raised from 78.12% to 82.48%. Another interesting finding is that further increasing the generalist data to 50k merely brings small improvements (from 82.48% to 84.0%). The results together with our experiments with LIMA suggest that adding a small number of generalist

data is sufficient to improve the specialist performance.

**Cross-instruction Robustness.** In all previous experiments, the models are trained and tested using the same instructions for each dataset. Now, we assess the model's robustness when confronted with alternative instructions that have not appeared during training. To do so, we employ ChatGPT (OpenAI, 2022) to generate 20 semantically equivalent instructions based on the original instruction. Figure 4 reports the results of these unseen instructions. As seen, the models trained with the addition of generalist data exhibit substantial improvement in average accuracy compared to the models trained with specialist data only. For instance, when the specialist data is limited to 2k instances, incorporating generalist data leads to a 6.64% absolute improvement on average compared to the specialist model. In the meantime, the incorporation of generalist data also alleviates the performance variation between the best-performing and worse-performing runs from 4.04% to 2.42%.

## 5 Experiments II: The Required Skills of the Target Tasks

We hypothesize that the model's ability to perform specific NLP tasks can be attributed to the mix of several core capabilities. As such, we set up three target tasks that focus on three key skills which are detailed below.

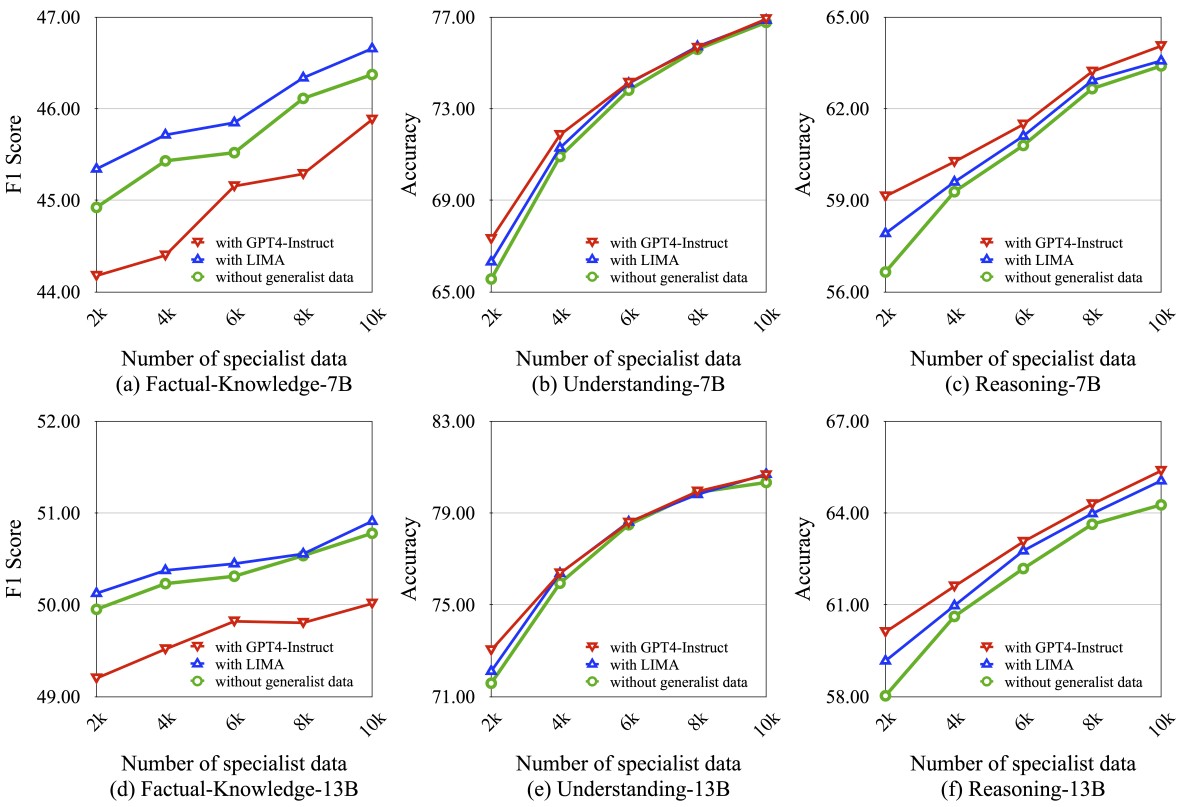

Figure 5: Comparison of models trained with different combinations of specialist and generalist data across different tasks. We report F1 score for Factual Knowledge and accuracy for other levels.

| Task Coverage | GPT4-Instruct | LIMA | specialist |
|---|---|---|---|
| Factual Knowledge | 14.78 | 17.28 | 46.37 |
| Understanding | 35.10 | 30.82 | 76.77 |
| Reasoning | 28.02 | 26.58 | 63.40 |

Table 2: The performance of generalists and specialists on tasks focusing on different skills. The specialists are trained with 10k task-specific instances on LLaMA-7B. For Factual Knowledge, the performance is measured by F1 score, while the others are measured by accuracy.

## 5.1 Skill Taxonomy

**Factual Knowledge** is essential for models to serve information needs. We use three knowledge-intensive datasets: Natural Questions (Kwiatkowski et al., 2019), WebQuestions (Berant et al., 2013), and TriviaQA (Joshi et al., 2017). All these three datasets consist of entity-centric questions, making them suitable for probing models' ability to activate and utilize factual knowledge. Following previous work (Brown et al., 2020), we evaluate under the closed-book setting where models are required to answer questions without the help of any external knowledge grounding.

**Understanding** acts as an important perspective as the capability to interpret input text. We choose the RACE dataset (Lai et al., 2017). RACE comprises data collected from English examinations in China and is specifically designed to assess the model's ability to read and deeply comprehend texts in real-world scenarios.

**Reasoning** is another fundamental ability for models to solve complex tasks. We use the SNLI (Bowman et al., 2015) dataset for implicit reasoning, the CQA (Talmor et al., 2019) for commonsense reasoning, and the AQUA (Ling et al., 2017) dataset for arithmetic reasoning.

## 5.2 Evaluation Setup

For the Factual Knowledge tasks, we use greedy search with a maximum generation length of 512. We adopt the F1 score as the evaluation metric following (Brown et al., 2020). For the Understating and Reasoning tasks, we utilize the same *classification with options* method detailed in Section 3.2 and report the model accuracy.

### 5.3 Results and Analysis

**Generalist models lag behind specialist models across all task skills.** Similar to Experiment I, we commence by comparing specialist and generalist models across three target tasks, each concentrating on distinct skills. The outcomes presented in Table 2 indicate that the generalist models consistently underperform the specialist models. For the Factual Knowledge task, the specialist model outperforms the generalist model with a 29.09 points higher F1 score. For the Understanding task, the specialist model surpasses the generalist model with a 41.67% increase in accuracy. For the Reasoning task, the specialist model excels beyond the generalist model, attaining a 35.38% absolute accuracy difference. Collectively, these findings substantiate the necessity of specialist tuning for accomplishing specific tasks.

**Incorporating GPT4-Instruct impairs the model's factual knowledge, while integrating LIMA offers benefits.** As illustrated in Figure 5, we observe the varying impact of different generalist data on the model's performance in the Factual Knowledge task. In particular, when GPT4-Instruct is incorporated, the F1 score experiences a decline. Conversely, when LIMA data is integrated, the F1 witnesses an increase. We argue that this difference stems from the fact that GPT4-Instruct is machine-generated, while LIMA is human-authored. The rationale is that machine-generated data may contain hallucinations, thus impairing the model's ability to recall factual knowledge.

To validate our hypothesis, we conduct experiments using additional generalist datasets, namely Dolly (Labs, 2023), and Evol-Instruct (Xu et al., 2023). Dolly consists of manually curated data generated by Databricks employees. Evol-Instruct uses more complex instructions than GPT4-Instruct and collects responses from ChatGPT (Fang et al., 2023). As observed in Figure 6, adding Dolly does not impair the performance, but incorporating Evol-Instruct leads to similar performance degradation as GPT4-Instruct. The above results are consistent with our hypothesis that machine-generated generalist data might adversely affect the model's factual knowledgeability due to hallucinations.

For a more rigorous comparison, we use Chat-GPT to generate responses for the 1k instructions in LIMA. The new 1k instruction-response pairs form a new generalist dataset, which we call LIMA-

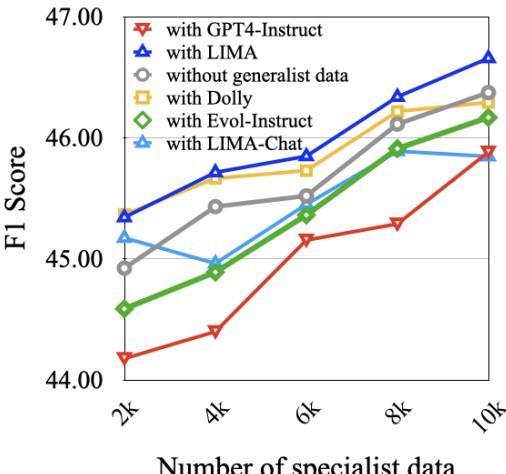

Figure 6: Results on the Factual Knowledge with LLaMA-7B.

Chat. The only difference between LIMA-Chat and LIMA is that the responses in LIMA-Chat are machine-generated, while those in LIMA are human-written. From Figure 6, we can see that LIMA-Chat indeed harms the performance while LIMA improves the performance. The above results suggest that the choice of generalist data is crucial for target tasks that heavily rely on factual knowledge.

**Adding generalist data enhances the understanding ability.** The results of the Understanding task are presented in Figure 5. It is evident that the addition of GPT4-Instruct greatly improves the model's performance when the specialist data is only 2k or 4k instances. However, as the number of specialist data further increases, the improvement diminishes. This suggests that the inclusion of generalist data can enhance the model's comprehension ability when the specialist data is limited.

**Adding generalist data enhances the reasoning ability.** We further evaluate the consequences of incorporating generalist data on the model's reasoning ability, as demonstrated in Figure 5. Notably, unlike Understanding, where the improvements from adding generalist data gradually diminish, the benefits of incorporating generalist data on the Reasoning tasks are persistent across different amounts of specialist data (an average improvement of 0.65% on LLaMA-7B and 1.12% on LLaMA-13B). This phenomenon could be attributed to the fact that the activation of reasoning

capabilities relies on diverse instruction data, and specialist data can be too narrow to fully unlock the true potential of LLMs.

**Effect of Model Scale.** For Factual Knowledge, increasing the model size from 7B to 13B results in more substantial performance improvements compared to increasing the amount of specialist data. This observation aligns with previous work (Brown et al., 2020), which indicates that an LLM's knowledge is mostly obtained through its pre-training. For Understanding, increasing the model size is as beneficial as adding more specialist data. For Reasoning, increasing the model size does not yield improvements as noticeable as Factual Knowledge and Understanding. We speculate that the emergence of strong reasoning abilities requires a larger model scale (Wei et al., 2022b).

**Generalist data plays a vital role in enhancing a model's understanding and reasoning capabilities, thereby increasing its effectiveness in addressing task-specific objectives.** We dissect the model's capabilities into three core components: (i) factual knowledge, (ii) understanding, and (iii) reasoning abilities. We demonstrate that incorporating generalist data does not improve the model's factual knowledge and, in some cases, may even be detrimental if it includes hallucinated information. Nevertheless, comparative experiments focusing on understanding and reasoning abilities reveal that generalist data effectively fosters the model's comprehension and significantly augments its reasoning capabilities.

This observed efficacy can be ascribed to the capacity of generalist data to facilitate the model's understanding and execution of diverse tasks. The wide range of instructions embedded within the generalist data stimulates the model's comprehension and reasoning faculties, empowering it to grasp specific requirements associated with various tasks more effectively. Moreover, by activating the model's reasoning abilities, it showcases enhanced performance across an assortment of tasks involving different levels of complexity.

The activation of comprehension and reasoning abilities further broadens the model's cognitive capacity, allowing it to derive a more comprehensive understanding based on existing information pertinent to the given task. Consequently, the inclusion of generalist data amplifies the model's task-specific capabilities, as it becomes adept at utilizing its expanded cognitive capacity to achieve superior performance.

# 6 Conclusions

In this study, we thoroughly investigated the interaction between specialist data and generalist data in the context of targeting specific NLP tasks. Our findings consistently demonstrate that the addition of generalist data leads to performance improvement when the task coverage is broad. This highlights the potential benefits of incorporating generalist data, particularly when the availability of specialist data is limited. Furthermore, we extensively examined the impact of integrating generalist data on the model's core capabilities. Surprisingly, we observed that the inclusion of generalist data did not enhance the model's factuality. In fact, generalist data containing hallucinatory information can have a negative impact. On the other hand, our experiments also revealed that the introduction of generalist data has positive effects on the model's understanding and reasoning abilities. Overall, our findings highlight the importance of leveraging generalist data to enhance the understanding and reasoning capabilities of NLP models, thereby enabling them to tackle various tasks more effectively. However, careful consideration should be given to the quality and reliability of the generalist data to avoid adverse effects on the model's factual knowledge.

# Limitations

While this work aims to provide a comprehensive investigation, we note that we do not exhaustively cover all possible evaluations. For example, we do not discuss NLP tasks such as summarization, translation, etc. Instead, we focus on constructing a hierarchy of four target tasks of different coverage levels and three target tasks focusing on different core skills. In addition, due to resource constraints, we only use LLaMA 7B/13B as our foundation models. We leave the investigation on different types and/or larger scales of models to our future work.

# Acknowledgments

This work was partly supported by the National Key Research and Development Program of China (No.2020YFB1708200), and the Shenzhen Science and Technology Program (JSGG20220831110203007).

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

# A   Instruction Template

============ INSTRUCTION FORMAT ============

Below is an instruction that describes a task, paired with an input that provides further context. Write a response that appropriately completes the request.

###Instruction:
[Task Prompt]

###Input:
[Input Text]

###Response:
[Output Text]