# OpenReview forum: "Specialist or Generalist? Instruction Tuning for Specific NLP Tasks"
_EMNLP/2023/Conference — EMNLP 2023 Main_

### Official Review · Reviewer_JSaf · 2023-07-27

**Soundness:** 3

**Excitement:**

4: Strong: This paper deepens the understanding of some phenomenon or lowers the barriers to an existing research direction.

**Paper Topic And Main Contributions:**

This paper is about instruction tuning. Specifically, they tune three models (backbone of Llama-7b) with "generalist" data consisting of GPT4-Instruct, LIMA, and no generalist data. Once the models are obtained, they instruct-tune the three models with "specialist" data and show an analysis across three target tasks on how much specialist data is needed to make an LLM a "specialist". They experiment with several data sizes for specialist data.

**Questions For The Authors:**

- Did you do an analysis of the overlap between instructions of the generalist data? If so, could you have combined GPT4-Instruct and LIMA for one generalist model? Especially with the findings for Table 1 vs 2 suggest that LIMA/GPT4-Instruct could be beneficial for different tasks.
- I'm missing some motivation for the 50k specialist training examples as in some cases the model doesn't seem to converge, e.g., Figure 1a, b. Why did you stop at 50k and not continue till convergence?
- Do you foresee any issues with the different sizes of the generalist data? E.g., GPT4-Instruct is 50k instances and LIMA is 1k.

**Reasons To Accept:**

- The analysis is insightful, I especially like the wide array of tasks and experiments covered and there is substantial evidence that the findings are robust across tasks.
- The paper is furthermore well written and follows a straightforward structure.

**Reasons To Reject:**

- I like the general idea of the analysis, though I feel the insights are not substantial enough for a long paper. For example, insights into the sizes of generalist data are lacking. What are the implications of performance on target tasks with different sizes of generalist data? Figure 3 gives some very brief insights, but the paper would improve given some extra insights for generalist data. I give several suggestions for making more space for in the section "Typos Grammar Style And Presentation Improvements".
- Overall, I feel the paper shows a rather trivial finding: More generalist + specialist data is better. It doesn't serve insights into the limitations of generalist/specialist data (i.e., convergence) and is not mentioned in the Limitations section.
- The authors mention a systematic guide in their contributions. However, there are insufficient details in the paper of such a guide. There is a list of findings from the analysis, but it does not give a systematic guide.

**Reproducibility:**

3: Could reproduce the results with some difficulty. The settings of parameters are underspecified or subjectively determined; the training/evaluation data are not widely available.

**Reviewer Confidence:**

2: Willing to defend my evaluation, but it is fairly likely that I missed some details, didn't understand some central points, or can't be sure about the novelty of the work.

**Typos Grammar Style And Presentation Improvements:**

Suggestions to make more space:
- Table 1 and Figure 1 can be merged if the figure started from 0 on the x-axis. In addition, Fig1+Table1 could have been squeezed into a 1x4 grid instead of 2x2, which then saves space.
- Since Fig 2,3,4 are used for Classification, you could squeeze them together again using subfigures which saves more space. Also, Table 2 can again be integrated in Fig 5, 6, 7 and squeezed together.


Other:
- Personal preference: The paper could have potentially started with broader coverage, e.g., starting with an analysis of the impact of scaling up the generalist data, finding convergence, and using that model to then further improve using specialist data. From Fig3, it doesn't seem that the model converged yet, and it's not too hard to increase the data to 100k (unless I'm missing something here).
- Some inconsistency with adding a period after a paragraph header.
- Dolly and Evol-Instruct do not seem integrated in Table 2.
- I'm surprised that the conclusion doesn't give any takeaways for the usage of specialist data and only mentions generalist data, would be good to add.

---

> ### Author Rebuttal · Authors · 2023-08-29
>
> **To Review #3**
>
> Thank you for your insightful comments and questions! We would like to provide our answers as follows:
>
> **R1: I like the general idea of the analysis, though I feel the insights are not substantial enough for a long paper. For example, insights into the sizes of generalist data are lacking. What are the implications of performance on target tasks with different sizes of generalist data? Figure 3 gives some very brief insights, but the paper would improve given some extra insights for generalist data. I give several suggestions for making more space for in the section "Typos Grammar Style And Presentation Improvements".**
>
> In Figure 3, we study the effect of the amount of generalist data. The results show that adding a small number of generalist data is sufficient to improve the specialist performance (the improvement is small after we increase the generalist data to 10k). In other experiments, comparing GPT4-Instruct (50k instances) and LIMA (1k instances), we can also see that their performance is close in many cases, indicating that the quantity is not the main factor, and we should better consider the quality and diversity. This is also in line with recent studies on generalist instruction tuning (Zhou et al., 2023; Gudibande et al., 2023) (also mentioned in Line 116-123).
>
> **R2+R3: Overall, I feel the paper shows a rather trivial finding: More generalist + specialist data is better. It doesn't serve insights into the limitations of generalist/specialist data (i.e., convergence) and is not mentioned in the Limitations section. The authors mention a systematic guide in their contributions. However, there are insufficient details in the paper of such a guide. There is a list of findings from the analysis, but it does not give a systematic guide.**
>
> Thank you for the question. We apologize for any confusion regarding the systematic guide. We summarize the key findings of our work as below:
> 1. Incorporating generalist data leads to performance improvements in NLP tasks, regardless of the task's coverage level.
> 2. Integrating generalist data is particularly beneficial when specialist data is limited.
> 3. Adding a small number of generalist data is sufficient to improve the specialist performance.
> 4. Adding generalist data enhances cross-task generalization and cross-instruction robustness.
> 5. The inclusion of generalist data does not necessarily enhance the model's factuality and may have negative effects.
> 6. Generalist data plays a vital role in enhancing a model’s understanding and reasoning capabilities, thereby increasing its effectiveness in addressing task-specific objectives.
> We consider that the above findings provide meaningful insights into when and how generalist data help specialist instruction tuning.
>
> **Q1: Did you do an analysis of the overlap between instructions of the generalist data? If so, could you have combined GPT4-Instruct and LIMA for one generalist model? Especially with the findings for Table 1 vs 2 suggest that LIMA/GPT4-Instruct could be beneficial for different tasks.**
>
> We deliberately choose the two generalist data sets, GPT4-Instruct and LIMA, for they vary significantly in many aspects, including the number of examples, the quality of data, the source of instructions, and the source of responses, etc. Specifically,  GPT4-Instruct contains 52k unique instruction-response pairs, where the instructions are collected through self-instruct and the responses are generated by GPT-4. LIMA consists of 1k carefully curated instruction-response pairs derived from human-authored community questions and answers. We emphasize that GPT4-Instruct serves as an example of generalist data synthesized by LLMs and LIMA represents an example of generalist data written by humans. Comparing the experiment results of the two datasets leads to interesting findings like the one you mentioned and also many others. For example, incorporating GPT4-Instruct  impairs the model’s factual knowledge while integrating LIMA offers benefits. The results give us insights that careful consideration should be given to the reliability of the generalist data to avoid adverse effects on the model’s factual knowledge. Your suggestion on combining the different generalist datasets is very insightful. It is an example that our research may inspire interesting future research. We are happy to add it in the next version. However, please kindly note that the focus of this paper is not to create better generalist data but a comprehensive evaluation using existing generalist data.
>
> **Q2: I'm missing some motivation for the 50k specialist training examples as in some cases the model doesn't seem to converge, e.g., Figure 1a, b. Why did you stop at 50k and not continue till convergence?**
>
>
> Thank you for the question, we are pleased to make the following classification. The primary focus of our experiments is to study the impact of generalists. Therefore, we are especially interested in the performance improvements brought by the generalist data rather than the absolute performance. Generally, the improvement from generalist data diminishes as the number of specialist data increases. From Figure 1 a and b, we can see that the improvement of generalist data already converges when the number is 50K. That is why we stop at 50K.
>
> **Q3: Do you foresee any issues with the different sizes of the generalist data? E.g., GPT4-Instruct is 50k instances and LIMA is 1k.**
>
> We deliberately choose the two generalist data sets, GPT4-Instruct and LIMA, for they vary significantly in the number of examples, the quality of data, the source of instructions, and the source of responses, etc. According to recent work (Zhou et al., 2023; Gudibande et al., 2023) , the quantity of generalist data is not a key factor. For example, LIMA (1k) has been found to outperform Alpaca (52k). AlpaGasus (9k), a subset of Alpaca outperforms Alpaca (52k). In the context of our study, we also find that in many cases the performances of LIMA and GPT4-Instruct are close, despite that Alpaca being 50x larger. Regarding the exclusive effect of data quantity, we present an analysis in Figure 3.
>
> **Presentation Issues**
>
> **P1: Suggestions for Space Optimization:**
>
> Your recommendations for merging tables and figures, as well as restructuring the layout, are much appreciated. Should the paper be accepted, we will indeed make the necessary revisions to improve the layout.
>
> **P2: Scaling Up Generalist Data:**
>
> Since the GPT4-Instruct data has a maximum of only 52k instances, it was not feasible to scale it to 100k. However, it is worth noting that the curve from 10k to 50k already conveys a clear trend, demonstrating that even a small amount of generalist data is effective. While more data might be better, the improvement seems marginal.
>
> **P3: Integration of Dolly and Evol-Instruct in Table 2:**
>
> We will ensure that the final version includes the results for Dolly and Evol-Instruct. Despite this omission, we want to emphasize that all generalist models lag behind specialist models.
>
> **P4: Conclusion and Specialist Data Usage:**
>
> Thank you for your constructive feedback. While the usage of specialist data for specific tasks is aligned with common understanding (i.e., more data for a particular task leads to better performance). Our work suggests to practitioners that they can reduce the demand for specialist data by including generalist data according to the nature of the target task.
>
>
> Once again, thank you for your constructive questions. We will include our answers in the next version.

---

### Official Review · Reviewer_b21v · 2023-08-03

**Soundness:** 3

**Excitement:**

4: Strong: This paper deepens the understanding of some phenomenon or lowers the barriers to an existing research direction.

**Paper Topic And Main Contributions:**

This work investigates the training benefits of using "generalist" and "specialist" datasets for  instruction tuning, comparing the relative benefits of each in various combinations.

**Questions For The Authors:**

It feels a little awkward to me to consider S-NI a "specialist" dataset. While I understand what the authors are referring to (the dataset is composed of fixed tasks with examples for each task), it is worth noting that S-NI has in aggregate more tasks than there are examples in LIMA. Though a secondary factor is that S-NI is probably less diverse in tasks than LIMA, I think the authors should more strongly justify why they consider S-NI a specialist dataset and how that fits into the broader message of their work.

Secondly, there are insufficient details on how the fine-tuning is performed (e.g. GPT4-Instruct and LIMA have very different number of examples: how is this balanced / what hyperparameters were used), and the exact format of combined fine-tuning on the generalist and specialist datasets are not clearly specified, unless I overlooked a section.

**Reasons To Accept:**

This work addresses a long investigated and still important question in transfer learning, which is more prominent in the present with the increasing popularity of "generalist" instruction tuning datasets. The datasets chosen actively used in research and for tuning of current highly capable models. The experiments run the mixture of training datasets are of the right format to answer these questions.

**Reasons To Reject:**

In general, I find Experiments II to be much weaker than Experiment I. The authors use a single task for each attribute they are examining, and appear to draw far too strong conclusions for the experiments they have run. A similar issue of over-extrapolation is found in the section L464-495, where there is a hypothesis that the reason for the difference in performance across datasets is due to being human-authored vs. machine-generated, but they supplement this experiment with a single experiment on an additional pair of datasets. Given that all four training sets are highly esoteric and greatly different, I do not believe there is any basis to make this claim at this point, and I would recommend the authors heavily soften the claim. (I acknowledge that the authors are already fairly conservative in their wording ("hypothesize", "are consistent with", "suggests"), but I nevertheless think that they need to water it down further.)

**Reproducibility:**

3: Could reproduce the results with some difficulty. The settings of parameters are underspecified or subjectively determined; the training/evaluation data are not widely available.

**Reviewer Confidence:**

4: Quite sure. I tried to check the important points carefully. It's unlikely, though conceivable, that I missed something that should affect my ratings.

---

> ### Author Rebuttal · Authors · 2023-08-29
>
> **To Review #2**
>
>
> We appreciate your discerning remarks and positive views regarding our work. Your thoughtful feedback is highly valued, and we are committed to addressing the questions you have raised.
>
> **R1: In general, I find Experiments II to be much weaker than Experiment I. The authors use a single task for each attribute they are examining, and appear to draw far too strong conclusions for the experiments they have run.**
>
> There seems to be some misunderstanding regarding the setups in Experiments II. For Factual Knowledge, we use a mixture of three datasets: Natural Questions, WebQuestions, and TriviaQA. For Reasoning, we also use a mixture of three datasets: SNLI (implicit reasoning), CQA (commonsense reasoning), and AQUA (arithmetic reasoning). Nevertheless, we understand that evaluating on these public NLP datasets might not completely reflect the knowledge, understanding, and reasoning abilities of LLMs. We will revise our writing accordingly to make our conclusions more task-specific.
>
> **R2: A similar issue of over-extrapolation is found in the section L464-495, where there is a hypothesis that the reason for the difference in performance across datasets is due to being human-authored vs. machine-generated, but they supplement this experiment with a single experiment on an additional pair of datasets. Given that all four training sets are highly esoteric and greatly different, I do not believe there is any basis to make this claim at this point, and I would recommend the authors heavily soften the claim. (I acknowledge that the authors are already fairly conservative in their wording ("hypothesize", "are consistent with", "suggests"), but I nevertheless think that they need to water it down further.)**
>
> Thanks for your insightful suggestion. Our experiments are based on two human-authored and two machine-generated generalist datasets. We understand that there are many other factors comparing different generalists and will explicitly add the limitations you mentioned.
>
> **Q1: It feels a little awkward to me to consider S-NI a "specialist" dataset. While I understand what the authors are referring to (the dataset is composed of fixed tasks with examples for each task), it is worth noting that S-NI has in aggregate more tasks than there are examples in LIMA. Though a secondary factor is that S-NI is probably less diverse in tasks than LIMA, I think the authors should more strongly justify why they consider S-NI a specialist dataset and how that fits into the broader message of their work.**
>
> Thank you for your insightful observation. Your comment brings attention to the distinction we are trying to make between SuperNI (S-NI) and LIMA, and we understand why this classification might initially seem ambiguous.
>
> The categorization of SuperNI as a "specialist" dataset in our work relies on the specific nature of the tasks within it. Although SuperNI encompasses a broad array of tasks (768 tasks are adopted in our experiments), these tasks are fundamentally geared towards natural language processing tasks distilled from real-world demands, such as determining different types of textual relationships, causality judgments of two sentences, selecting the best answers, summarization, translation, etc. They focus on fixed and definable tasks that can often have an approximate standard answer.
> Here's an illustrative example from SuperNI:
>
> \===================================================================
>
> **Example from SuperNI:**
>
> Task: In this task you're given two statements. You must judge whether the second sentence is the cause or effect of the first one. Label the instances as “cause” or “effect” based on your judgment. The sentences are separated by a newline character.
> Input: The women met for coffee. They wanted to catch up with each other.
> Answer: cause.
> \===================================================================
>
> Contrast this with an example from LIMA, which may encompass mathematical concept explanations, lifestyle advice, geographical information, personal experiences, etc. The instructions within LIMA are more oriented towards user needs and are not as easily captured by conventional single tasks.
>
> \===================================================================
>
> **Example from LIMA:**
>
> Input: How to be a lazy college student?
> Answer: In college, unlike in high school, you are expected to take responsibility for your own learning. It is ultimately up to you to determine how much time and effort you want to put into your schoolwork. Experts suggest that college students today work considerably fewer hours than their counterparts did in 1961. But there is also evidence that working more hours is not necessarily the best way to be productive. A little laziness may even be good for us. Part 1 of 3: prioritize what’s important…
>
> \===================================================================
>
> Our classification of SuperNI as a specialist dataset emphasizes the task-specific nature of the challenges it poses, while LIMA's generalist nature accommodates a broader, user-oriented spectrum of inquiries.
>
> I hope this explanation addresses your concern and provides the justification for our categorization of SuperNI. Another point is that the SuperNI dataset is used as specialist data in the broader context of Experiments I, where we deliberately construct a variety of target tasks with varying scopes of coverage (i.e., how specific the target task is).
>
> **Q2: Secondly, there are insufficient details on how the fine-tuning is performed (e.g. GPT4-Instruct and LIMA have very different number of examples: how is this balanced / what hyperparameters were used), and the exact format of combined fine-tuning on the generalist and specialist datasets are not clearly specified, unless I overlooked a section.**
>
> We deliberately choose the two generalist data sets, GPT4-Instruct and LIMA, for they vary significantly in the number of examples, the quality of data, the source of instructions, and the source of responses, etc. Throughout all experiments, we stick to a common training strategy: as stated in line 231-239, we fine-tune the model using the Adam optimizer with learning rate 2e-5 for 3 epochs. Additionally, we use a batch size = 128.
>
> For the data format, as stated in Line 212-217, we follow the template used in Stanford’s Alpaca project for all generalist and specialist datasets.

---

### Official Review · Reviewer_ReUf · 2023-08-03

**Soundness:** 4

**Excitement:**

4: Strong: This paper deepens the understanding of some phenomenon or lowers the barriers to an existing research direction.

**Paper Topic And Main Contributions:**

This paper proposes to examine whether general purpose instruction tuning contributes to building specialized models, across a variety of tasks. They show task performance improves with more specialized training data (as expected) and that generalist data generally improves task performance even with lots of specialized data (although general purpose data is not enough).

TLDR summary: I like this paper and learned from it, which I think is the sign of a good paper. However, there are some slight deficiencies in the experimental setting. Overall, I have a favorable impression.

**Questions For The Authors:**

From the weaknesses:
1. Can you explain the reasoning behind the model choices?
2. Can you explain the dataset choices and why not more tasks?

**Reasons To Accept:**

- The paper is timely and relevant to much of the NLP community which is adapting to instruction data and LLMs
- The work is relatively easy to read and includes lots of great analysis
- The analysis includes several instruction datasets that are used today (Lima, GPT-instruct, etc.) and a large number of experiments

**Reasons To Reject:**

- The analysis is only performed with one model. Although the one model is timely and relevant, it would be nice to see this replicated with another model (or even with different tuning, like LoRA).
- Classification, sentiment, and Yelp are very similar tasks IMO. Perhaps my fellow reviewers would disagree, but I would like to see something more different like QA or summarization (which are in SuperNI, so I’m not sure where those fit in this paper?)
- There aren’t any generalist datasets used that are published that I can find (are Lima and GPT4-Instruct published?) and although they are well hyped and (perhaps?) well used now, it’s hard to assess their quality. It would have been nice to see some more established generalist fine-tuning datasets like FLAN or those used in T0, etc. The authors may have had a good reason for this, looking forward to the rebuttal.

Some of these may be a miscommunication, I'm open to increasing my score in Soundness from 3 to 4 on the dataset/modeling issues.

**Reproducibility:**

4: Could mostly reproduce the results, but there may be some variation because of sample variance or minor variations in their interpretation of the protocol or method.

**Reviewer Confidence:**

4: Quite sure. I tried to check the important points carefully. It's unlikely, though conceivable, that I missed something that should affect my ratings.

**Typos Grammar Style And Presentation Improvements:**

I think a label before the legend in Figures like Figure 1 would be helpful (something like “General Data Used” or something. It took me a bit to parse the figures, but I think they are well done once you figure them out.

It would be nice to have somewhere a longer description of these generalist datasets. I’m not familiar with evol-instruct for example. Perhaps this could go in the appendix? This doesn’t influence my score, but just a suggestion.

---

> ### Author Rebuttal · Authors · 2023-08-29
>
> **To Review #1**
>
> Thank you for your valuable insights and favorable impression of our work. We appreciate your thoughtful feedback, and we would like to address your concerns as follows:
>
> **R1:The analysis is only performed with one model. Although the one model is timely and relevant, it would be nice to see this replicated with another model (or even with different tuning, like LoRA.**
>
> Thanks for your valuable suggestion! The main reason for us doing experiments with only one model (Llama-7B) is the budget of computational resources. We choose Llama-7B for its great popularity in recent works and moderate computational cost. We are actively conducting experiments on larger models (e.g., Llama-13B) with the more efficient training method, LoRA, as you suggested. For the numbers we have collected, we anticipate similar findings regarding the impact of generalist data.
>
> **R2: Classification, sentiment, and Yelp are very similar tasks IMO. Perhaps my fellow reviewers would disagree, but I would like to see something more different like QA or summarization (which are in SuperNI, so I’m not sure where those fit in this paper?))**
>
> We appreciate your perspective, and we'd like to explain our rationale for this choice. One primary purpose of Experiments I is to assess a variety of target tasks with distinct levels of generality (i.e., how specific the target task is). We construct a hierarchy of four specialist tasks: SuperNI (multiple tasks, multiple formats) > Classification (multiple tasks, single format) > Sentiment (single task, multiple domains) > Yelp (single task, single domain). Specifically, the Classification task encompasses a wide variety of challenges (e.g., cause-effect classification, sentence composition, text matching, toxic language detection, ethics classification, spam classification, etc) that are NOT included in the Sentiment tasks. The Sentiment task encompasses multiple domains (e.g., food/movie/ book/music reviews, Twitter posts, Reddit posts, etc), while the Yelp task only covers product/service reviews from the Yelp platform.
> The reason for choosing classification tasks over QA and summarization to construct the above hierarchy is that QA and summarization are much more difficult to evaluate. For example, different QA and summarization datasets have different output formats (e.g., some are phrases while others are complete sentences). In our preliminary experiments, we found that this issue has a significant impact on the final evaluation result. However, we acknowledge that presenting an extra experiment on QA and summarization would be beneficial and will add it in the next version.
>
> **R3: There aren’t any generalist datasets used that are published that I can find (are Lima and GPT4-Instruct published?) and although they are well hyped and (perhaps?) well used now, it’s hard to assess their quality. It would have been nice to see some more established generalist fine-tuning datasets like FLAN or those used in T0, etc. The authors may have had a good reason for this, looking forward to the rebuttal.**
>
> Thank you for sharing your concerns regarding the generalist datasets used in our work, specifically LIMA and GPT4-Instruct. It's entirely understandable to seek clarity on these datasets, and we are pleased to provide some context.
> You can find the datasets for LIMA and GPT4-Instruct at the following links:
> LIMA: https://huggingface.co/datasets/GAIR/lima
> GPT4-Instruct: https://huggingface.co/datasets/vicgalle/alpaca-gpt4
>
> Regarding your mention of more established datasets like FLAN or T0. We would like to clarify that, in our work's context, FLAN, T0, and SuperNI are considered as specialist data, because they are constructed from existing NLP datasets focusing on specific tasks and fixed instruction templates. Note that SuperNI has already been used in our experiments. We only consider datasets like LIMA and GPT4-Instruct fit the category of generalist data in our study as they aim to meet any real user demands.
>
> **Q1: Can you explain the reasoning behind the model choices?**
>
> Thanks for your thoughtful question. Please refer to our answer to R1.
>
> **Q2. Can you explain the dataset choices and why not more tasks?**
>
> Thanks for your thoughtful question. Please refer to our answer to R2.
>
> **P1. Presentation Improvements**
>
> We appreciate your feedback on the presentation and will incorporate your suggestions to enhance the clarity of the paper. Thank you for your guidance.

---

### Meta-Review · Area_Chair_24fu · 2023-09-19

**Recommendation:** 5

**Metareview:**

The reviewers are positive about the soundness of the paper. The work is timely, addresses an important question in transfer learning, and the analysis is thorough. The writing is easy to understand. The reviewers raised some issues about the experimental design, some of which the authors addressed during the author response period.

Overall, the reviewers believe the paper is sound and are excited to see it published.

---

### Decision · Program_Chairs · 2023-10-07

**Decision:**

Accept-Main

**Comment:**

The reviewers are positive about the soundness of the paper. The work is timely, addresses an important question in transfer learning, and the analysis is thorough. The writing is easy to understand. The reviewers raised some issues about the experimental design, some of which the authors addressed during the author response period.

Overall, the reviewers believe the paper is sound and are excited to see it published.